# A Feasible Approach to Developing Fiber-Enriched Bread Using Pomegranate Peel Powder: Assessing Its Nutritional Composition and Glycemic Index

**DOI:** 10.3390/foods12142798

**Published:** 2023-07-23

**Authors:** Paula García, Andrés Bustamante, Francisca Echeverría, Cristian Encina, Manuel Palma, Leyla Sanhueza, Verónica Sambra, Maria Elsa Pando, Paula Jiménez

**Affiliations:** 1Departamento de Nutrición, Facultad de Medicina, Universidad de Chile, Santiago 8380453, Chile; pgarcia@uchile.cl (P.G.); anbustama@uchile.cl (A.B.); vero.sambrav@uchile.cl (V.S.); pandosanmartin@uchile.cl (M.E.P.); 2Carrera de Nutrición y Dietética, Departamento Ciencias de la Salud, Facultad de Medicina, Pontificia Universidad Católica de Chile, Santiago 7820436, Chile; franciscaecheverria@uc.cl; 3Nutrición y Dietética, Facultad de Ciencias de la Salud, Universidad Autónoma de Chile, Santiago 7500912, Chile; cristian.encina@uautonoma.cl; 4P&M Foods, Los Olmos 3465, Santiago 7810668, Chile; manueljpalma@gmail.com; 5Departamento Ciencia de los Alimentos y Tecnología Química, Facultad de Ciencias Químicas y Farmacéuticas, Universidad de Chile, Santiago 8380492, Chile; leyla.sanhueza@ug.uchile.cl

**Keywords:** bread, pomegranate peel, agro-industrial by-products, dietary fiber, glycemic index, glycemic response

## Abstract

The consumption of dietary fiber (DF) has been associated with a reduced incidence of non-communicable diseases. Despite various strategies implemented worldwide to increase DF intake, it remains low. Therefore, the development of new fiber-rich food products that are widely consumed could be a strategy to improve DF intake. In this study, an agro-industrial by-product, pomegranate peel powder (PPP), was used as an innovative source of DF and antioxidant. The objective was to develop a bread enriched with DF, antioxidants, and sensory characteristics by partially replacing wheat flour (WF) with PPP at levels of 0%, 2.5%, 5%, 7.5%, and 10%. Bread with 2.5% and 5% PPP was chosen for a clinical trial to evaluate glycemic response (GR) in healthy subjects and determine the bread’s glycemic index (GI). As the percentage of PPP increased, both the DF and total polyphenol content increased significantly. The highest overall acceptability was achieved with bread containing up to 5% PPP. Consumption of bread with 2.5% and 5.0% PPP significantly reduced the GI compared to the control bread, while the decrease in GR was not significant. PPP could be a potential food and low-cost ingredient to improve the bread’s nutritional quality through its contribution to DF and antioxidants.

## 1. Introduction

Cardiovascular diseases are the leading cause of death in middle-income countries and affect both men and women worldwide. Obesity, hypertension, dyslipidemia, and diabetes are the main risk factors for these diseases [1]. A healthy lifestyle and dietary pattern are crucial in reducing these risk factors. The intake of dietary fiber (DF) has been shown to prevent overweight and obesity [2], metabolic syndrome [3], as well as the incidence and mortality of cardiovascular diseases [4]. The Institute of Medicine (2005) [5] established adequate DF intake levels of 25 and 38 g/d for young women and men, respectively, based on their cardiovascular health effects. Special attention has been paid to soluble dietary fiber (SDF), which is fermented by bacteria that produce short-chain fatty acids responsible for health effects associated with a decrease in the prevalence of chronic diseases, such as type 2 diabetes and coronary heart disease, and a reduction in risk factors, including a decrease in postprandial blood glucose [6]. Its intake has also been associated with improved cholesterol levels and inflammatory markers [7]. Despite the implementation of various programs or strategies to increase fiber intake in most countries, its dietary intake is still inadequate. Therefore, developing new fiber-rich food products is necessary to increase the general population’s DF intake.

Bread is a commonly consumed bakery product worldwide and is considered a suitable vehicle for fortification with various nutritional and non-nutritional compounds [8]. However, bread is typically made from refined (white) wheat flour (WF) and water, resulting in low fiber content, high carbohydrate content, and often high glycemic index. Therefore, efforts have been made to develop healthier types of bread, such as whole grain or multigrain bread [9,10]. Nevertheless, these foods mainly provide insoluble dietary fiber (IDF), are not widely appreciated, and often come with higher prices [11]. Other studies have reported the use of corn, soybean, and bean by-products to improve the nutritional profile (the content of proteins, fibers, and phenolic compounds, among others) in gluten-free bakery products. However, there may be a positive or negative effect on the technological properties (color, texture, volume, porosity, thickness, homogeneity, etc.) and mainly on the sensory characteristics depending on the percentage added [12].

A novel source of DF and bioactive compounds is provided by some agro-industrial by-products [13]. These residues are a current issue because they produce greenhouse gas emissions, land and water pollution, and a misuse of resources, which have a negative impact on society, the economy, and the environment. In this context, the 2030 agenda for the United Nations sustainable development goals (SDGs) set food waste reduction targets (SDG 12) [14]. Due to this, several countries have adopted strategies to move toward a circular economy [15]. On the other hand, these wastes have very low-cost raw materials whose valorization could allow for the development of added-value products with environmental and technological advantages [16]. For example, pomegranate (*Punica granatum* L.) is a fruit that belongs to the family *Punicaceae*, which is native to Central Asia and whose cultivation has spread throughout the Mediterranean basin and the Americas with a global production of 3 million tons [17]. It is frequently used for juice production, leaving peels and seeds as by-products of processing in proportions of 73% and 27%, respectively [18]. Pomegranate peel (PP) could be considered a valuable residue since it has a high DF (~28–40%) [19] and phenolic compound content, mainly ellagitannins, such as punicalagin [20]. Thus, the incorporation of PP in bread formulations would allow for the development of fiber-rich bread by using a low-cost raw material.

Various studies have incorporated pomegranate peel powder (PPP) into experimental bread formulations [21,22,23,24,25,26,27]. Typically, wheat bread has been fortified with PPP within the range of 1–7.5% [22,23,24,26,28]. However, other authors have tested higher levels of PPP addition, reaching up to 10–18% [21,22,27]. The fortification of bread with PPP led to a significant increase in fiber content. For example, Mehder et al. (2013) [24] observed a 3.6-fold increase in crude fiber when 5% of PPP was added, resulting in a total fiber content of 4.82% in the bread. Tharsini and Sangwan (2018) [26] reported a 1.5-fold increase in crude fiber when incorporating 6% of PPP in the bread formulation. Abolila et al. (2019) [21] described a substantial increase in crude fiber (7.4-fold increase) upon fortifying bread with 18% PPP, reaching a crude fiber content of 7.29% in the final product. However, all these authors determined crude fiber and did not measure DF, SDF, and IDF.

Overall, the acceptability of bread fortified with PPP was higher at addition levels that did not exceed 5%. For instance, Sayed-Ahmed (2014) [23] reported that bread fortified with 2.5% and 5% PPP received higher scores in overall acceptability and physical properties when compared to control bread and the 7.5% treatment. Similarly, based on sensory evaluation, Palak et al. (2020) [27] found that bread with 5% PPP treatment scored better than the 10% and 15% treatments in terms of color and appearance, body and texture, taste and flavor, and overall acceptability of the product. In parallel, the addition of PPP to bread can potentially have a substantial impact on the dough’s characteristics. For instance, Abolila et al. (2019) [21] demonstrated that PPP-fortified loaves exhibited a lower loaf volume compared to the control sample, with a reduction of 21.4% in bread containing 18% PPP. 

Furthermore, there is scarce information regarding the glycemic response caused by the consumption of foods supplemented with PPP. A double-blind, randomized, placebo-controlled study was conducted on 22 individuals with type 2 diabetes to evaluate the effect of the daily consumption for 8 weeks of bread elaborated with the incorporation of PPP (1%) on metabolic and biochemical parameters. The study observed decreases in serum insulin, triglyceride, and total cholesterol levels in the individuals in the treatment group compared with those in the control group. Nevertheless, this study only evaluated the effect of bread consumption on fasting blood glucose and not the acute effect on postprandial blood glucose [29]. Additionally, no studies have reported the determination of the glycemic index (GI) of the bread formulated with PPP and the glycemic response (GR) in subjects after its consumption. Therefore, this study was designed to gain new insight into this topic. The main objective was to develop a prototype of bread enriched with DF, antioxidants, and adequate sensory characteristics by partially replacing wheat flour with PPP at different percentages. Furthermore, a clinical trial was conducted to evaluate the GR in healthy subjects and to determine the GI of the bread. The information generated by this study seeks to provide valuable information to the bakery industry and health professionals regarding the development of new DF-rich foods.

## 2. Materials and Methods

### 2.1. Pomegranate Material Recovery

PP (cv. Wonderful) was obtained as a byproduct of the fruit from a commercial vineyard located in Maule Region, Chile. The PP was dried using convection in an air-drying tunnel (without a brand, built with a Tetlak motor), with a horizontal airflow rate of 2 m/s and 50% of recirculation at 50 °C for 48 h. The resulting dried product was ground in a knife mill (Polymix® System PX-MFC 90 D, Kinematica AG, Malters, Switzerland) to obtain PPP with a particle size of 20 mesh and stored in a dark, room-temperature environment.

### 2.2. Characterization of PPP

#### 2.2.1. Proximate Analysis and Determination of DF

The moisture content, total protein, and ash were determined according to the official methods of the A.O.A.C. (2005) [30]. Lipids and carbohydrates were determined by using acid hydrolysis and Anthrone methods, respectively. The gravimetric enzymatic method [26] was used to determine total dietary fiber (TDF), SDF, and IDF. All analyses were performed in triplicate.

#### 2.2.2. Determination of Total Phenolic Content (TPC)

Phenolic compounds were extracted from PPP through solid–liquid extraction using ethanol: water (40:60 *v*/*v*) for 3 h at 159 rpm. The mixture was then centrifuged at 3000 rpm for 4 min at room temperature. The TPC was quantified spectrophotometrically using a Folin–Ciocalteu phenol reagent assay [31]. The absorbance of samples was measured at 765 nm, and the results are expressed as milligrams of gallic acid equivalents per gram of PP in dry weight (mg GAE/g DW) based on a calibration curve (150–750 mg GAE/L, R2: 0.9986). All analyses were performed in triplicate.

#### 2.2.3. Determination of Punicalagin Content

To determine punicalagin, PPP was extracted as described previously using solid–liquid extraction with ethanol:water (40:60 *v*/*v*). Punicalagin was detected and quantified using high-performance liquid chromatography (HPLC) with a Merck Hitachi L-6200 pump, a Waters 996 photodiode-array detector (DAD), and a C18 column (5 µm, 4.6 mm i.d. × 250 mm, Symmetry, Waters, Ireland), following the method described by Zhang et al. (2009) [32] with some modifications. To provide a brief description of the HPLC method, solvent A (0.4% aqueous phosphoric acid) and solvent B (acetonitrile) were used as the mobile phases in a multistep gradient: 0 min (5% B); 10 min (15% B); 30 min (25% B); 35 min (5% B). The sample injection volume was 20 µL and the flow rate was 1.0 mL/min at room temperature. The monitored wavelength was 360 nm, and the results are expressed as milligrams of punicalagin per gram of pomegranate peel in DW (mg/g DW), based on a calibration curve ranging from 12 to 200 mg punicalagin/L extract (R2: 0.9942). All analyses were conducted in triplicate.

#### 2.2.4. Determination of the Antioxidant Capacity (AC)

The Ferric Reducing Antioxidant Power (FRAP) assay was conducted following the method of Benzie and Strain (1996) [33] with some modifications. The absorbance was measured at 593 nm, and the results are expressed as mmol FeSO4/g DW. All analyses were performed in triplicate.

The free radical 2,2-diphenyl-1-picrylhydrazyl (DPPH) assays were carried out according to the procedure of Bondet et al. (1997) [34]. The absorbance of the sample was measured using a UV absorption spectrophotometer at 517 nm, and the results are expressed as EC50 (mg/mL). Each analysis was performed in triplicate.

#### 2.2.5. PPP Shelf-Life Study

For the shelf-life tests, 250 g of PPP was placed in open paper bags inside an oven (Memmert UF-75, Buechenbach, Germany) set at 30 °C for a period of 9 months. PPP samples were collected at defined time intervals, with one sample per month. The following analyses were carried out in the samples: TPC, microbiological analyses (Salmonella spp., filamentous fungi, and yeasts) according to the method described by Andrews et al. (2023) [35], and color parameters (L*, a*, b*, C*, and h°) with a HunterLab Spectrophotometer UltraScan PRO (Reston, VA, USA).

### 2.3. Preparation of Bread Formulations Incorporating PPP 

The bread was prepared using WF (1000–900 g), water (500 mL), yeast (20 g), sugar (5 g), and salt (10 g) as ingredients, and various percentages of PPP (0, 2.5, 5.0, 7.5, and 10.0 g per 100 g of WF) for the replacement of WF. These percentages were determined based on previous trials and data from the literature. 

#### 2.3.1. Characterization of Bread Formulations

##### Rheological properties of Bread Doughs and Physical Properties of Baked Bread 

To determine the rheological properties of bread doughs, an alveograph (Alveolab graph, Chopin Technology, France) was used. The following parameters were measured: flour strength (W), resistance to extension (tenacity (P)), and dough extensibility (L). To determine the specific volume (cm^3^/g) of baked bread, the loaf bread volume was divided by the loaf bread weight.

##### Proximate Analysis and Determination of DF

The moisture content, total protein, ash, lipid, TDF, SDF, and IDF analyses were performed using A.O.A.C. (2005) [30] methods. Carbohydrates were determined by difference. All analyses were conducted in triplicate. 

##### Determination of TPC

To determine the TPC of the bread formulations, polyphenols were extracted from the baked bread. One gram of bread was added to a mixture of methanol and water (80:20 *v*/*v*) acidified with 0.1% HCl. The mixture was stirred at 150 rpm for 2 h and then centrifuged for 5 min. The supernatant was separated and used for determinations. TPC was determined using a colorimetric assay with Folin–Ciocalteu phenol reagent, following the method of Singleton and Rossi (1965) [31]. All analyses were performed in triplicate.

##### Microbiological Analysis 

A filamentous fungi test was assayed by using the methods of Andrews et al. (2023) [35]. 

##### Sensorial Analysis

A sensory evaluation of overall acceptability was conducted on bread samples within 24 h of baking. The samples were assessed by a panel of at least 60 untrained consumers (regular bread consumers) using the hedonic scale method, with scores ranging from 1 to 7 (7 = extremely like, 6 = very much like, 5 = like, 4 = neither like nor dislike, 3 = dislike, 2 = dislike very much, 1 = dislike extremely). The results were obtained by calculating the overall mean. The acceptability percentage was calculated as the ratio of the number of tests with acceptability scores ≥5 to the total number of tests.

### 2.4. Clinical Assay

#### Calculation of Glycemic Index of Bread 

Subjects: Nine healthy adults (six women, three men) between 27 and 44 years old, with a body mass index (BMI) between 18.5 and 24.9 kg/m², who had maintained their body weight for the last six months, were selected. Anthropometric evaluations, including weight and height, were conducted using standardized methods. Only healthy individuals who did not consume any medication or supplements were considered. Those with diagnosed underlying pathologies, allergies, or intolerances to test foods, and women with polycystic ovary syndrome were excluded from the study. All subjects signed an informed consent form (No. 178-2022) approved by the Ethics Committee of the Human Beings Research Center (CEISH) of the Faculty of Medicine at the University of Chile.

Experimental design: The protocol used in this study was based on the FAO guidelines (1998) [36]. Volunteers were instructed to maintain their usual diet throughout the study period. The tests were administered only to individuals who met the following conditions in the 24 h prior to each test: regular dietary habits, abstinence from intense physical exercise, and abstinence from alcohol and tobacco. The subjects were instructed to fast for ten hours prior to each session. The participants were instructed to remain seated during the test and were not allowed to consume water or any other food.

Test foods: This study utilized bread formulations with 2.5% and 5% PPP incorporation, as they were rated highest in the sensory evaluation. The control food used to measure the GI was bread made with wheat flour without PPP. The amount used for the tests was based on 50 g of available CHO.

Glycemic response (GR): At the start of each intervention, two fasting capillary blood samples were taken at time 0 min. The average of these values was considered as the baseline blood glucose concentration, which should be <100 mg/dL to proceed with the next stage of the session. Subsequently, capillary blood samples were obtained at 15, 30, 45, 60, 90, and 120 min after bread consumption. The samples were collected by finger puncture (capillary sample), and glycemia was measured using the Accu-Chek^®^ Instant glucometer.

Calculation of the Glycemic Index (GI): The area under the curve (AUC) was calculated geometrically for each food using 50 g of available CHO. The area under the baseline (fasting glycemia) was excluded, using the trapezoidal rule. White bread (glycemic index of 100) was used as the control. The GI of bread with PPP 2.5% and 5% consumed by each subject is expressed as the ratio between the area under the test food curve/area under the curve of white bread (control) × 100. To obtain the final value of the GI (average of the GIs obtained in each test food), the values were classified into low glycemic index (≤55), medium glycemic index (56–69), and high glycemic index (≥70) [37].

### 2.5. Statistical Analysis

The statistical analyses were conducted using a one-way or multifactor ANOVA test to compare means, depending on the case. When significant differences were found, the Tukey HSD (honest significant differences) multiple-comparison test (*p* ≤ 0.05) was applied. Statgraphics Centurion XV (Version 15.1.02, StatPoint, Inc., Warrenton, VA, USA) was used for the analyses. The normal distribution of variables for GI was verified using the Shapiro–Wilk test. The results of the measured variables are expressed as mean ± standard deviation (SD). Repeated measures ANOVA was used to compare the GR and GI of breads. All analyses considered *p* < 0.05 to be significant.

## 3. Results and Discussion

### 3.1. Characterization of PPP

#### 3.1.1. Proximate Analysis and DF 

The composition of the PPP was as follows: 6.7 ± 0.05% moisture, 5.0 ± 0.10% protein, 4.2 ± 0.02% ash, 3.6 ± 0.03% fat, and 30 ± 0.8% available carbohydrates (Table 1). In general, the literature reports a wide range of values for proximate analyses of PP from different varieties, where moisture fluctuated between 9.3 and 13.7%, protein from 0.7 to 5.8%, ash from 2.7 to 6.0%, and fat from 0.4 to 6.5% [21,23,24,38,39,40,41,42,43]. For example, Akuru et al. (2020) [38] informed similar values for moisture (6.7%) and ash (4.1%) contents in Wonderful pomegranate, with lower protein content (2.2%) and higher fat level (6.5%).

In this study, PPP had dietary fiber (DF) values of 50.5%, 13.3%, and 37.2% for TDF, SDF, and IDF, respectively. PPP was found to contain a higher DF content than pomegranate fruit (50% vs. 18%). Studies on pomegranate varieties other than Wonderful have reported DF values ranging from 31% to 66% [44,45]. For PP of the Wonderful variety, some authors have reported lower values than this study with 43.5%, 35.3%, and 8.2% to TDF, IDF, and SDF, respectively [42,43]. According to these results, it could be established that PP from the Wonderful variety had a high TDF content within the range described in the literature. Genotype and environmental factors have been found to significantly contribute to the TDF content in vegetables, cereals, and leguminous foods [44]. Based on the daily recommended allowances (RDA) of DF in adults, PPP could be considered a good source of TDF [19] and a potential healthy ingredient for food development.

#### 3.1.2. Total Phenolic and Punicalagin Content

The study results showed that the TPC was 172.4 ± 1.7 mg GAE/g DW (Table 1), which was higher than that reported by Galaz et al. (2017) [46] who obtained a TPC value of 107.8 mg GAE/g DW for PPP of the Wonderful variety using a drum drying process and conventional methanol 80% extraction. Similarly, García et al. (2021) [47] described a TPC of 125 mg GAE /g DW by using the air-drying tunnel process and extracting with ethanol: water by pressurized liquid extraction. Conversely, Akuru et al. (2020) [38] reported a lower TPC (143 mg GAE/ g DW) by drying PP in an oven and extracting it with ethanol, while a higher TPC (432.7 mg GAE/g DW) was obtained using solid–liquid extraction with ethanol: water (20:80 *v*/*v*) for the Wonderful variety [48]. In general, pomegranate fruit is rich in phenolic compounds, with the peel containing the highest amount of fruit total phenolic content, particularly hydrolysable tannins such as punicalagin and ellagic acid, compared to other parts of the fruit [49]. Therefore, it is possible that these compounds are responsible for the high TPC observed in this study. 

In this research, PPP showed a punicalagin content of 80 mg/g DW. Previous studies have reported punicalagin values in PP ranging from 1.6 to 476 mg/g DW [48,50,51,52,53,54]. Rongai et al. (2017) [48] reported a higher value (216.8 mg/g DW) for the Wonderful variety than this study (80 mg/g DW) by using ethanol:water (20:80 *v*/*v*). In contrast, using the supercritical CO_2_ extraction method with ethanol as a cosolvent, Bustamante et al. (2017) [50] found a value of 97 mg/ g DW, while García et al. (2021) [47] reported a lower content (17.6 mg/g DW) by using ethanol:water with pressurized liquid extraction technology.

#### 3.1.3. Antioxidant Capacity

The DPPH free radical and FRAP assays are commonly used to evaluate the AC of natural products. The radical-scavenging activity of DPPH was calculated as EC50, which corresponds to the concentration of the extract (mg/mL) required to inhibit 50% of the initial DPPH free radical. In this study, PPP showed a higher AC value for DPPH (EC50 of 0.05 mg/mL) than those reported by Sharayei et al. (2019) [55] for aqueous extracts (ranging from 0.2 to 1.2 mg/mL) and by Kennas and Amellal-Chibane (2019) [56] for ethanolic and acetone extracts (0.076 and 0.16 mg/mL, respectively). In contrast, Elfalleh et al. (2012) [57] reported a higher AC with EC50 values of 0.0011 and 0.0038 mg/mL for an aqueous and methanolic extract of PP, respectively. 

Regarding FRAP, the value obtained in this study (1430 µmol Fe + 2/g DW) falls within the range of 287–1950 µmol Fe + 2/g DW described for PP of cv. ShisheKape-Ferdos [55]. It is worth noting that the variations in the values of TPC, punicalagin content, and AC of PPP obtained in this study, as compared to those reported in the literature, may be attributed to various agronomic characteristics of the fruit, such as genotype, variety, growing region, or climate, as well as the pre-treatment of the raw material and the extraction and quantification methods used.

### 3.2. PPP Shelf-Life Study (TPC Stability, Microbiological Quality, and Color)

The TPC stability of PPP was monitored during 9 months of storage at 30 °C (Table 2). It was observed that the TPC decreased significantly from the beginning of the study (172.4 mg GAE/g DW) until the second month (121 mg GAE/g DW). Later, the TPC remained stable until the 7th month without significant changes. From the 8th month until the 9th month, there was a significant increase in TPC, reaching a value of 177.4 mg GAE/g DW. The study on the stability of these compounds showed similar behavior to that found by several studies on fruits, which have reported that polyphenol content decreases with increasing drying temperatures due to possible thermal degradation of the antioxidants [58,59]. In a PPP storage stability test at 4 °C for 3 months, Çam et al. (2014) [59] reported a significant decrease in TPC content during the first 15 days compared to the rest of the storage period, suggesting that phenolic compounds on the surface of the powder could be more exposed to oxidation. Also, the different degrees of susceptibility of PPP polyphenols to oxidation could contribute to variable oxidation stability during storage. However, TPC remained stable until the 7th month without significant changes, possible due to the thermal inactivation of hydrolytic and oxidative enzymes caused by the drying process, thus preventing a greater loss of polyphenols [46]. The increase in TPC from the 8th month until the 9th month could be explained by the hydrolysis of polymerized compounds with high molecular weight, such as punicalagin, into compounds of lower molecular weight [60]. The degradation of ellagitannins during storage has also been described to give rise to ellagic acid and lower-molecular-weight compounds, which may impact TPC [61]. Similarly, Robert et al. (2010) [62] reported an increase in polyphenol and anthocyanin retention during the storage at 60 °C for 56 days of encapsulated juice and ethanolic extract of pomegranate, possibly due to the hydrolysis of pomegranate-conjugated polyphenols.

On the other hand, the results showed that PPP had a complete absence of Salmonella spp. throughout the evaluation period, as well as filamentous fungi and yeast counts of up to 5 × 102 CFU/g, which complies with the Chilean Food Legislation [63] for WF due to the absence of a specific microbiological regulation for PPP.

The color of the PPP samples remained consistent (*p* < 0.05) throughout the 9-month shelf-life study. The color-plotting diagrams provided the following Cartesian coordinates, a* (14.3 ± 1.8), b* (51.2 ± 3.6), L* (43.2 ± 2.6), c* (53.2 ± 3.9), and h° (74.5 ± 3.9), indicating that PPP maintained a consistently reddish-brown color during storage.

### 3.3. Characterization of Bread Formulations

#### 3.3.1. Proximate Analysis and DF of the Bread

The results of the proximate chemical analysis, including moisture, ash, protein, fat, carbohydrate, and dietary fiber of the bread, are presented in Table 1. The moisture content increased significantly as a higher percentage of PPP was incorporated into the bread formulation. It was apparent that the high fiber content of PPP increased water retention during the bread formulation process. Conversely, protein and carbohydrate content decreased with a higher PPP content. This could be attributed to the lower carbohydrate content of PPP compared to WF. The data collected in this study showed similar behavior to those reported by Sayed-Ahmed (2014) [23] and Mehder (2013) [24] when incorporating PPP into bread at levels ranging from 2.5% to 5%. However, these authors obtained slightly higher values for proteins (12.5% to 12.9%) and carbohydrates (79.5% to 80.4%). In contrast, the fat content (2.8% to 5.3%) was significantly higher in their study, whereas it did not vary significantly among the different formulations in our study. The variations in the results could be attributed to several factors, including the type of pomegranates used to extract PPP, the process followed for bread formulations, and the specific ingredients utilized, such as butter and oil.

As expected, the incorporation of PPP resulted in a direct increase in TDF, SDF, and IDF. However, these results cannot be directly compared to other studies that have investigated the incorporation of PPP at different percentages for bread formulations, as those studies primarily focused on determining crude fiber content [23,24]. Tharshini and Sangwan (2018) [26] characterized various bread formulations prepared with WF and soybean flour, with a constant soybean flour content of 10% and partial replacement of WF with PPP ranging from 2% to 6%. The study found that SDF and IDF significantly increased with higher PPP incorporation compared to the control bread. The values for SDF rose from 1.27% to 2.26%, while IDF values increased from 6.13% to 7.01%. These values were lower than those observed in our study with the incorporation of 2.5% and 5% PPP. Similar trends in the chemical composition of bread have been reported by other authors who have evaluated the incorporation of PPP into various bakery products. For instance, biscuits with 7.5% PPP showed an increase in TDF from 3.65% to 5.52% [45]. In our study, with the same level of PPP incorporation into bread, the increase in TDF was from 3.7% to 12.1%.

It is important to note that in this research, we utilized the var. Wonderful pomegranate. However, the other articles discussed did not mention the specific pomegranate variety used for obtaining PPP. It is worth mentioning that PPP derived from different pomegranate cultivars exhibit diverse chemical compositions, TDF content, and IDF/SDF ratios [44]. Therefore, when comparing results, this aspect should be taken into consideration. Additionally, the type of baked product and the elaboration process should also be considered.

#### 3.3.2. Rheological Parameters of the Bread Doughs and Physical Properties of Bread

The flour strength, known as W, exhibited a decrease from 178 × 10^−4^ J for WF to 39 × 10^−4^ J for WF + 10% PPP. This reduction in strength was also observed in the dough extensibility, denoted as L, which decreased from 50 mm to 8 mm. Moreover, the specific volume of the bread decreased from 2.3 cm^3^/g for WF to 1.8 cm^3^/g for WF + 10% PPP. In contrast, the dough resistance to deformation, represented by P, increased from 99 mm for WF to 152 mm for WF + 10% PPP. However, while the incorporation of PPP in the bread increased the TDF, SDF, IDF, and polyphenol content, it also resulted in changes in the rheological properties of the bread dough and the physical properties of the bread. When flour comes into contact with water, it forms gluten. Therefore, a higher protein content in the flour leads to the formation of more gluten. Gluten is an elastic protein that helps the dough retain its shape and trap gas during fermentation. Consequently, a higher W value indicates greater strength and the ability of the dough to withstand the pressure from fermentation gas, resulting in bread with a larger volume. The results of this study demonstrated a decrease in the W value, which can be attributed to the incorporation of PPP in the bread formulations, leading to a reduction in wheat gluten content (dilution effect) [22]. Additionally, the network of gluten is physically disrupted by the fibers from PPP [64]. Furthermore, the specific volume of the bread (volume of the loaf bread divided by its weight) decreased, which could be attributed to the higher content of PPP. A higher percentage of PPP likely increased the weight of the bread, possibly due to its higher water retention resulting from the high fiber content of PPP. Additionally, the dilution effect on gluten and the slower formation of the gluten network may have contributed to this decrease [21]. Alterations in the rheological properties of doughs and the physical properties of baked goods elaborated with PPP, such as bread [21,22,23,24], biscuits [43,44,45,46,47,48,49,50,51,52,53,54,55,56,57,58,59,60,61,62,63,64,65], and muffins [42], have also been reported in previous studies.

#### 3.3.3. Total Phenolic Compound

The TPC values for the control bread, 2.5%, 5%, 7.5%, and 10% PPP are presented in Table 1. As anticipated, there was a significant increase in TPC as the percentage of PPP in the bread formulation increased, owing to the contribution of polyphenols present in the PPP.

In terms of the TPC of bread, Sayed-Ahmed (2014) [23] reported higher values for bread formulations with 0%, 2.5%, 5%, and 7.5% PPP, reaching values of 0.8, 2.3, 3.5, and 5.4 mg GAE/g DW, respectively. These variations could be attributed to differences in the pomegranate variety, bread formulations, the extraction process of polyphenols, and the quantification method employed, among other factors.

#### 3.3.4. Microbiological and Sensory Analyses

The filamentous fungi count was determined for the different bread formulations (n = 5) in accordance with the requirements of the Chilean Food Legislation [63]. Overall, all bread samples were found to be microbiologically acceptable (<10^2^), and in compliance with national regulations (10^2^ to 10^3^ CFU/g).

In terms of sensory analysis, the results are consistent with those reported in the literature. Several studies suggest that incorporating up to 5% PPP in bread formulations, in general, does not negatively affect bread acceptability [22,23,25,26].

The sensory quality of a food product plays a vital role in determining its overall acceptability and consumers’ intention to purchase. In the present study, the results of the overall acceptability test revealed that the bread formulation with 2.5% PPP achieved the highest acceptability score (6.0 = liked very much), followed by the 5% PPP formulation and the control sample (5.9 = like) (Table 3). No statistically significant differences were observed among these three formulations. As anticipated, bread formulations with higher PPP content (7.5% and 10%) exhibited lower acceptability scores. Judges detected a residual sour and bitter taste, a dark color resembling whole bread (Figure 1), and a hard crust as the percentage of PPP increased in the bread formulation. These factors might have influenced the acceptability score and purchase intention. However, it is noteworthy that the samples containing 7.5% and 10% PPP obtained scores of 5.4 and 4.8, respectively, indicating a certain level of liking (5 = like) or indifference (4 = neither like nor dislike) rather than dislike. Consequently, all types of bread were considered organoleptically acceptable. A similar trend was observed in the percentages of acceptability and purchase intention, which declined as the PPP content increased.

### 3.4. Clinical Study

#### Glycemic Index of Bread and Glucose Response

The average blood glucose levels and incremental area under the curve did not exhibit significant differences among the various types of bread in the subjects included in the study (Figure 2B,C). The recorded values were 113 ± 17 mg/dL for white bread (control), 112 ± 15 mg/dL for bread with 2.5% PPP, and 109 ± 11 mg/dL for bread with 5% PPP. However, significant differences in the glycemic index (GI) were observed between the control bread and the bread formulations with PPP. The GI values were 100, 78, and 72 for control bread, bread with 2.5% PPP, and bread with 5% PPP, respectively (Figure 2A). No significant differences were found between the two PPP formulations.

The glycemic index (GI) is a measure of a food’s capacity to increase blood glucose levels, proposed by Jenkins et al. (1981) [66], and serves as an indicator of the quality of carbohydrates (CHO) in the diet. Food items are classified based on their GI as low GI (≤55), medium GI (56–69), or high GI (≥70). White bread is commonly used as the reference food with a GI value of 100. According to the latest update of GI tables by Atkinson et al. (2021) [67], bread could show a wide range of GI values (24 to 100), with one in three pieces of bread having a high GI. On average, bread from Asian countries had the highest GI (68 ± 16). Bread from Germany and Scandinavian countries, which are typically made from rye and other grains, may have lower GI values; nevertheless, limited information was available regarding their specific GI values.

In the present study, the incorporation of PPP into the bread formulations resulted in a decrease in glycemic response (GR) and GI. However, a significant decrease in GI was observed only between the control bread and the formulations with PPP. The lack of a significant effect of PPP on blood glucose levels in the current study could be explained by the small sample size [29], and the percentage of PPP added to the bread formulations. In this study, 2.5% and 5% of PPP were used, as higher concentrations rendered the bread less acceptable. Among the weaknesses of the study, we can mention that the insulin response or the insulinemic index was not analyzed. In addition, we did not assess the menstrual phase of the volunteers, which may affect insulin sensitivity.

High levels of fiber and phenolic compounds in PPP have been associated with improved glucose control. DF reduces or delays carbohydrate absorption, which helps inhibit increases in insulin levels, improve glycemic control [68], and reduce the GI of foods. In turn, consuming low-GI foods and/or following low-GI diets has been shown to produce a greater satiation and satiety effect [69,70], as they generate glycemic curves that are slower and more stable over time [71]. Therefore, the intake of dietary fiber is a factor associated with multiple health benefits, including the control of body weight [72]. An adequate intake of DF (25 g/day for women; 38 g/day for men) has been associated with a decreased risk of colorectal cancer, weight loss through appetite–satiety regulation, improvement in immune function through the interaction of fiber with the microbiota, and promotion of intestinal health [73]. Specifically, the intake of SDF has been associated with a decrease in the prevalence of NCDs, such as heart disease, cancer, chronic respiratory disease, and diabetes, and a reduction in risk factors by generating a decrease in blood pressure, inflammatory parameters, ultrasensitive C-reactive protein, cardiovascular risk markers, LDL-C concentration, and postprandial glycemia [74,75]. In this context, the consumption of a diet with a low GI and a greater contribution of DF can help reduce the risk of developing type 2 diabetes [76], a condition that affects 12.3% of the population over 15 years of age.

## 4. Future Prospects

Consumer awareness of the environmental and health advantages related to bread products incorporating non-wheat ingredients, expected to exhibit a lower glycemic index, is on the rise. Recent global events, such as the COVID-19 pandemic and the Russia–Ukraine conflict, have had enduring impacts on wheat production and the supply chain [77,78]. Thereby, the development of novel ingredients holds promise in mitigating pressure on food system resilience and bolstering food security. Diversifying plant-based food sources assumes paramount importance in enhancing sustainability within global food systems, as it not only reduces the environmental impact but also fosters local economic development [79]. A diversified food supply chain facilitates access to affordable and nutritious food, particularly during crises, therefore promoting food system resilience. Additionally, non-wheat raw materials offer compelling nutritional properties, encompassing health-promoting compounds like dietary fibers and antioxidants [80]. Despite limitations in technological quality and sensory attributes relative to wheat substitution products, the transformation of food byproducts into valuable resources presents a significant opportunity to transition toward a closed-loop economic system [77]. Consumer decisions in the food market are influenced by various factors, including taste, smell, appearance, texture, functional properties, and nutritional value. Factors perceived as positively impacting health, such as product origin and food technology enhancing health benefits, also play a crucial role in consumer decision making [81]. Considering the 2030 sustainable development goals, specifically goal 12 related to responsible production and consumption, addressing the existing knowledge gaps in nutritional, health, technological, and environmental dimensions through interdisciplinary approaches is imperative [82]. The synergy between human nutrition, food science, and sustainability is indispensable for scaling up research outcomes and achieving substantial positive impacts on a large scale [83]. Future research could aim to expand the area of study and/or the limits of the system where innovations can reduce the costs of inputs, also involving other stages of the supply chain, or evaluate consumer acceptance in the case of modifications of the recipe [78].

## 5. Conclusions

Nowadays, the revalorization of agro-industrial by-products is of great importance due to their environmental impact. In this research, the incorporation of PPP in bread formulations ranging from 2.5% to 10% led to an increase in their TDF content, including SDF and IDF, as well as antioxidants compared to the control bread. The highest overall sensory acceptability was achieved with PPP incorporation levels up to 5%. In the clinical trial, the addition of 2.5% and 5% PPP resulted in a decrease in the GR and GI, although a significant decrease was observed only for the GI. To achieve more significant reductions, higher levels of PPP incorporation and a larger number of healthy subjects in the clinical study may be required. Therefore, further studies are needed to improve the sensory characteristics of bread with PPP incorporation levels exceeding 5%. This would allow for increased dietary fiber intake and provide health benefits, such as better glycemic response.

## Figures and Tables

**Figure 1 foods-12-02798-f001:**
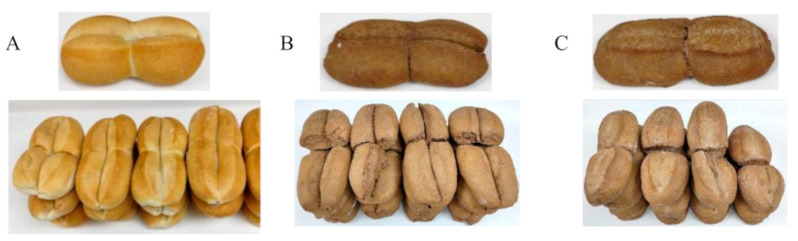
Bread formulations. (**A**) Control bread; (**B**) bread+ 2.5% PPP; (**C**) bread + 5% PPP.

**Figure 2 foods-12-02798-f002:**
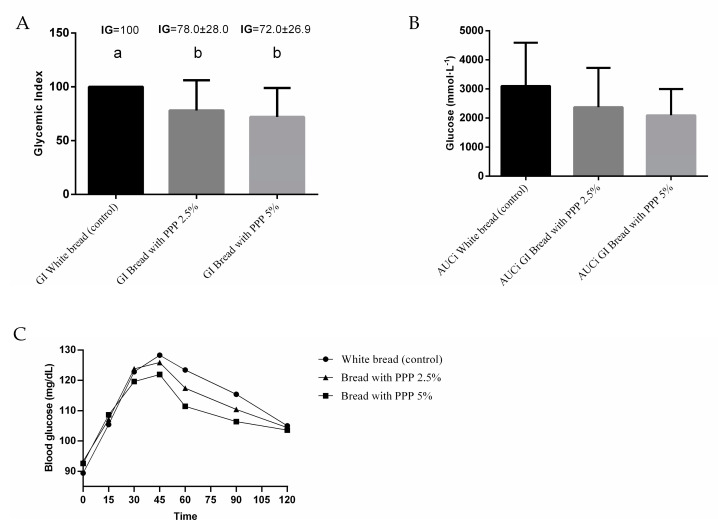
(**A**) Glycemic index: Values expressed as average ± SD (n = 9). Significant differences between tests are expressed as different superscript letters. Repeated measures ANOVA. Control vs. bread with PPP 5% (*p* = 0.014); control vs. bread with PPP 2.5% (*p* = 0.046); (**B**) glucose; AUCi (the incremental area under the curve); (**C**) average blood glucose responses.

**Table 1 foods-12-02798-t001:** Total phenolic content, proximal chemical analysis, and dietary fiber of PPP and bread elaborated with PPP incorporation.

	TPC	Moisture	Ash	Protein	Fat	Carbohydrate	DF		
							TDF	IDF	SDF
	mg GAE/g DW	g/100 g	g/100 g DW				g/100 g DW		
PPP	172.4 ± 1.7	6.7 ± 0.05 ^a^	4.2 ± 0.02 ^a^	5.0 ± 0.10 ^a^	3.6 ±0.03 ^a^	30.0 ± 0.8 ^a^	50.5 ± 0.9 ^a^	37 ± 0.9 ^a^	13.3 ± 0.9 ^a^
Control bread	0.5 ± 0.04	32.7 ± 0.1 ^b^	1.87 ± 0.07 ^b^	12.4 ± 0.04 ^c^	1.1 ± 0.2 ^b^	81,2 ± 0.7 ^b^	3.7 ± 0.1 ^b^	2.0 ± 0.0 ^b^	1.7 ± 0.0 ^b^
Bread 2.5% PPP	1.3 ± 0.2	32.1 ± 0.4 ^b^	1.5 ± 0.8 ^b^	11.9 ± 0.4 ^bc^	1.2 ± 0.2 ^b^	75.1 ± 1.1 ^b^	10.3 ± 0.9 ^c^	7.1 ± 0.8 ^c^	3.2 ± 0.3 ^c^
Bread 5.0% PPP	2.2 ± 0.1	34.0 ± 0.1 ^c^	2.0 ± 0.2 ^b^	11.7 ± 0.5 ^bc^	1.3 ± 0.2 ^b^	74.5 ± 1.0 ^b^	10.4 ± 1.2 ^c^	7.3 ± 1.0 ^c^	3.1 ± 0.1 ^c^
Bread 7.5% PPP	3.2 ± 0.1	34.4 ± 0.3 ^c^	2.50 ± 0.01 ^b^	11.4 ± 0.3 ^bc^	1.1 ± 0.2 ^b^	72.9 ± 0.3 ^b^	12.1 ± 0.9 ^d^	9.2 ± 1.0 ^d^	2.9 ± 0.5 ^c^
Bread 10% PPP	3.8 ± 0.06	36.1 ± 0.3 ^d^	2.78 ± 0.08 ^b^	10.8 ± 0.08 ^b^	1.2 ± 0.07 ^b^	70.5 ± 0.3 ^b^	14.7 ± 0.1 ^e^	10.5 ± 0.2 ^e^	4.2 ± 0.1 ^d^

PPP: Pomegranate peel powder, DW: dry weight, TPC: total polyphenol content, DF: dietary fiber, TDF: total dietary fiber, IDF: insoluble dietary fiber, SDF: soluble dietary fiber. Different letters in the same column indicate significant differences between samples (*p* ≤ 0.05).

**Table 2 foods-12-02798-t002:** Evolution of color, total phenolic content, and microbiological counts of pomegranate peel powder stored at 30 °C for 9 months.

Storage Period	Color					TPC	Microbiological Counts		
Months	L* (D_65_)	a*(D_65_)	b* (D_65_)	C* (D_65_)	h° (D_65_)	mg GAE/g DW	*Salmonella* spp. (CFU/g)	Filamentous Fungi (CFU/g)	Yeasts (CFU/g)
0	43.2 ± 2.6 ^a^	14.3 ± 1.8 ^a^	51.2 ± 3.6 ^a^	53.2 ± 3.9 ^a^	74.5 ± 0.9 ^a^	172.4 ± 1.7 ^a^	ND	˂10 ^a^	˂10 ^a^
1	43.6 ± 2.8 ^a^	13.2 ± 1.8 ^a^	49.0 ± 3.3 ^a^	50.8 ± 3.7 ^a^	74.9 ± 0.9 ^a^	127.4 ± 7.8 ^ab^	ND	˂10 ^a^	˂10 ^a^
2	43.3 ± 0.9 ^a^	15.7 ± 0.7 ^a^	54.7 ± 1.5 ^a^	56.9 ± 1.7 ^a^	74.0 ± 0.3 ^a^	121.0 ± 15.2 ^b^	ND	˂10 ^a^	˂10 ^a^
3	44.2 ± 0.7 ^a^	14.2 ± 0.5 ^a^	52.3 ± 1.0 ^a^	54.2 ± 1.0 ^a^	74.8 ± 0.5 ^a^	147.4 ± 6.0 ^b^	ND	˂10 ^a^	˂10 ^a^
4	44.9 ± 0.3 ^a^	15.0 ± 0.0 ^a^	55.4 ± 0.1 ^a^	57.4 ± 0.1 ^a^	74.8 ± 0.1 ^a^	139.5 ± 8.2 ^b^	ND	˂10 ^a^	˂10 ^a^
5	44.4 ± 1.3 ^a^	14.4 ± 1.4 ^a^	50.5 ± 5.0 ^a^	52.6 ± 5.2 ^a^	74.1 ± 0.1 ^a^	126.1 ± 0.0 ^b^	ND	˂10 ^a^	˂10 ^a^
6	43.9 ± 0.1 ^a^	14.9 ± 0.3 ^a^	52.6 ± 1.0 ^a^	54.6 ± 1.1 ^a^	74.2 ± 0.1 ^a^	129.4 ± 4.3 ^b^	ND	˂10 ^a^	˂10 ^a^
7	42.6 ± 0.7 ^a^	15.4 ± 0.4 ^a^	54.7 ± 0.4 ^a^	56.9 ± 0.5 ^a^	74.2 ± 0.3 ^a^	133.9 ± 5.2 ^bv^	ND	˂10 ^a^	˂10 ^a^
8	43.5 ± 0.9 ^a^	14.7 ± 0.8 ^a^	53.7 ± 0.9 ^a^	55.7± 2.3 ^a^	74.5 ± 0.7 ^a^	173.3 ± 8.5 ^a^	ND	˂10 ^a^	˂10 ^a^
9	44.5 ± 0.8 ^a^	14.7 ± 0.9 ^a^	53.7 ± 0.8 ^a^	54.9 ± 1.2 ^a^	74.9 ± 0.5 ^a^	177.4 ± 7.6 ^a^	ND	˂10 ^a^	5 × 10^2 b^

TPC: total phenolic content. ND: not detected. Different letters in the same column indicate significant differences between samples (*p* ≤ 0.05).

**Table 3 foods-12-02798-t003:** Overall acceptability of bread formulations with different proportions of PPP incorporation (%*w*/*w*).

Bread Formulation	Average Score	1–3 N (%)	4 N (%)	5–7 N (%)
Control bread	5.9 ± 0.9 ^a^	1.6	0	98.4
Bread 2.5% PPP	6.0 ± 0.7 ^a^	0	0	100
Bread 5.0% PPP	5.9 ± 0.7 ^a^	0	1.6	98.4
Bread 7.5% PPP	5.4 ± 0.9 ^b^	1.3	10.5	88.2
Bread 10% PPP	4.8 ± 1.2 ^c^	13.6	24.2	62.1

PPP: pomegranate peel powder. An acceptability score >4.0 indicates adequate acceptability of the product. The average score is presented as mean ± standard deviation. Different superscript letters indicate significant differences between bread formulations (*p* < 0.05).

## Data Availability

The data used to support the findings of this study can be made available by the corresponding author upon request.

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
