# Peer review of "A Feasible Approach to Developing Fiber-Enriched Bread Using Pomegranate Peel Powder: Assessing Its Nutritional Composition and Glycemic Index"

_foods, 2023, doi:10.3390/foods12142798_

Round 1

Reviewer 1 Report

Thank you for submitting the manuscript "A novel approach to developing fiber-enriched bread using pomegranate peel powder: Assessing its Nutritional Composition and Glycemic Index" to Foods. Overall, the study is interesting and relevant. However, I do have some suggestions for improving the manuscript.

Line#19: inappropriate is a word that is not appropriate. I even agree that the strategies are still not so successful, but they are adequate. They just didn't hit everyone.

I believe that if there really is a novel strategy it would be interesting to include it in the abstract. As described, the abstract is just another product development work partially replacing non-wheat flour.

Line#39: From the way this is written it seems that if dietary fiber is ingested in any amount it brings this benefit, and this is not true.

Line#52-57: the authors wrote a whole paragraph about the disadvantages of wheat flour, but it is necessary to consider that partial replacement does not eliminate these disadvantages and most of them are still present in the bread developed in the work.

Line#60: consider citing https://doi.org/10.1007/s11947-022-02975-1

Line#74-75: this is a sentence that does not support authors using the expression "new approach" in the title

Line#78-84: Consider including how much PPP was included in the bread and how much was given to the volunteers.

Line#153: per 100 g or 1000 g? If it is per 1000 g, the abstract as well as the whole text presented the percentages incorrectly.

Line#291 and # 425: It's not interesting to use abbreviations in titles.

Table 2: I believe that in place of a comma are points.

Line#460: Please consider placing the footnote below the table.

Figure 2: Can the area of the curve be calculated? It would be interesting to include.

English language is fine.

Author Response

Reviewer 1

1.- Line#19: inappropriate is a word that is not appropriate. I even agree that the strategies are still not so successful, but they are adequate. They just didn't hit everyone.

Answer: In the abstract (L18), the word “inadequate” was changed to low

2.- I believe that if there really is a novel strategy it would be interesting to include it in the abstract. As described, the abstract is just another product development work partially replacing non-wheat flour.

Answer: L20-21, the following sentence was rewritten: “In this study, was used an agro-industrial by-product, pomegranate peel powder (PPP), as an innovative source of DF and antioxidant”.

3.- Line#39: From the way this is written it seems that if dietary fiber is ingested in any amount it brings this benefit, and this is not true.

Answer: L40-42, it is established in the text that adequate DF intake levels of 25 and 38 g/d for young women and men, respectively, based on their cardiovascular health effects

4.- Line#52-57: the authors wrote a whole paragraph about the disadvantages of wheat flour, but it is necessary to consider that partial replacement does not eliminate these disadvantages and most of them are still present in the bread developed in the work.

Answer: According to the results of Table 1, it is observed that a higher percentage of PPP was incorporated into the bread formulation, increasing the content of dietary fiber, ash, and polyphenols, while decreasing the protein and carbohydrate content with respect to control bread. It is important to consider that the replacement of wheat flour must be partial, obtaining better technological characteristics (elasticity of the dough, bread volume) and sensory attributes at the 5% level, as also indicated in other studies. Higher percentages of incorporation of PPP affect these characteristics

5.- Line#60: consider citing https://doi.org/10.1007/s11947-022-02975-1

Answer: L57-62 Citation was considered in the text, as follows: “Other studies have reported the use of corn, soybean, and bean by-products to improve the nutritional profile (the content of proteins, fibers, and phenolic compounds, among others) in gluten-free bakery products. However, there may be a positive or negative effect on the technological properties (color, texture, volume, porosity, thickness, homogeneity, etc.) and mainly on the sensory characteristics depending on the percentage added”

6.- Line#74-75: this is a sentence that does not support authors using the expression "new approach" in the title:

Answer: The title was changed as follows: “A feasible approach to developing fiber-enriched bread using pomegranate peel powder: Assessing its Nutritional Composition and Glycemic Index”.

7.- Line#78-84: Consider including how much PPP was included in the bread and how much was given to the volunteers.

Answer: L105-L107 The following was added to the text: “22 individuals with type 2 diabetes and bread elaborated with the incorporation of PPP (1%)”

8.- Line#153: per 100 g or 1000 g? If it is per 1000 g, the abstract as well as the whole text presented the percentages incorrectly.

Answer: L183 In the text, it was modified to per 100g

9.- Line#291 and # 425: It's not interesting to use abbreviations in titles.

Answer: Line#322 AC was modified to Antioxidant Capacity

                Line#447 TPC was modified to Total Phenolic Compound

10.- Table 2: I believe that in place of a comma are points.

Answer: In values of Table 2, commas were changed to points             

11.- Line#460: Please consider placing the footnote below the table.

Answer: In Table 3, the footnote was placed (L512-L514)

12.- Figure 2: Can the area of the curve be calculated? It would be interesting to include

Answer: Yes, the area was calculated and considered in the discussion of the results. (Figure 2 The area under the curve incremental). This calculation is explained in lines 252-259 “Calculation of the Glycemic Index (GI): The area under the curve (AUC) was calculated geometrically for each food using 50 grams of available CHO. The area under the baseline (fasting glycemia) was excluded, using the trapezoidal rule. White bread (glycemic index of 100) was used as the control. The GI of bread with PPP 2.5% and 5% consumed by each subject was expressed as the ratio between the area under the test food curve/area under the curve of white bread (control) x 100. To obtain the final value of the GI (average of the GIs obtained in each test food), the values were classified into the low glycemic index (≤55), medium glycemic index (56-69), and high glycemic index (≥70)”

Reviewer 2 Report

Thank you for the opportunity to review the manuscript entitled “A novel approach to developing fiber-enriched bread using pomegranate peel powder: Assessing its Nutritional Composition and Glycemic Index”. The purpose of the research is to explore pomegranate peel as an innovative source of dietary fiber and antioxidant, as to develop a bread enriched with dietary fibers, antioxidants, and sensory characteristics by partially replacing wheat flour. 

The manuscript is original and investigated a topical and interesting issue. 

The abstract is clear and comprehensive. The authors introduce the context, the purpose of the research, the experiment method and the main insights. 

In the section “Introduction”, together with the health benefits, I would stress an important (and essential) aspect towards a fair, healthy and environmentally friendly food system. Considering that the bread suggested in the current research contains pomegranate peel, which is actually an agro-waste (or food waste, or by-product, as you prefer), I would stress its important also towards sustainability. The authors only cite such an aspect at L. 93, but I believe it should be developed further, both in the “Introduction”, in the “Literature review” and also in the “Discussions” (as suggested in the next comments). 

The section “Introduction” should highlight, at the end, to whom the results of the research are addressed (i.e., the audience of the research). 

A section addressed to (critically) investigate the “Literature review” is missing. I suggest the authors developing a semi-systematic or narrative review on the previous studies, which have explored the issue of substituting wheat flour with other components, always stressing both sustainability (economic, social, environmental) and healthy aspects. 

In the subsection “Calculation of glycemic index of bread”, how the nine healthy adults have been selected? Which sampling strategy was adopted? Please, add some more information. It nine adults a representative sample? I suggest the authors highlighting such biases/limitations in the research. 

The description of the “Results”, considering the methods adopted and the purpose of the research, is rather interesting and clear. 

I have some minor suggestions. In a suitable section addressed to “Discussion” or “Future research directions”, I would introduce the future analysis of the market opportunities of such a bread, also considering its environmental (and economic) benefits. Nowadays, due to the inflation rate and the Russia-Ukraine conflict, the price of bread has steadily increased reaching never-experienced prices. Could the (partial) substation of wheat flour with pomegranate peel powder both an environmental and an economic strategy? Please, add some discussions on the topic. 

In addition, future research directions could investigate consumers’ behavior towards their willingness to buy such a bread, as well as possible implications in terms of production costs. For your interest, you should make reference to studies analyzing the bread supply chain, as to open new paths to environmental and economic studies. In my opinion, it is important to highlight the nexus between eco-friendly friendly and healthy products, also in the direction of the Sustainable Development Goals of the United Nations. 

Drewnowski, A., Finley, J., Hess, J.M., Ingram, J., Miller, G., Peters, C. (2020). Toward Healthy Diets from Sustainable Food Systems. Current Developments in Nutrition, 4(6),083. https://doi.org/10.1093/cdn/nzaa083.

Amicarelli, V., Lombardi, M., Varese, E., Bux, C. (2023). Material flow and economic cost analysis of the Italian artisan bread production before and during the Russia-Ukraine conflict. Environmental Impact Assessment Review, 101, 107101. https://doi.org/10.1016/j.eiar.2023.107101

Brancoli, P., Bolton, K., Eriksson, M. (2020). Environmental impacts of waste management and valorisation pathways for surplus bread in Sweden. Waste Management, 117, 136-145. https://doi.org/10.1016/j.wasman.2020.07.043.

Author Response

Reviewer 2:

1.- The section “Introduction” should highlight, at the end, to whom the results of the research are addressed (i.e., the audience of the research).

Answer: The following sentence was added (lines 117-119): The information generated by this study seeks to provide valuable information to the bakery industry and health professionals regarding the development of new DF-rich foods.

  1. A section addressed to (critically) investigate the “Literature review” is missing. I suggest the authors developing a semi-systematic or narrative review on the previous studies, which have explored the issue of substituting wheat flour with other components, always stressing both sustainability (economic, social, environmental) and healthy aspects

Answer: To provide clarity on the antecedents, a paragraph was added (lines: 81-104)

Typically, wheat bread has been fortified with PPP within the range of 1-7.5% [22,23,24,26,28]. However, other authors have tested higher levels of PPP addition, reaching up to 10-18% [21, 22,26]. The fortification of bread with PPP led to a significant increase in dietary fiber (DF) content. For example, Mehder et al. (2013)[24] observed a 3.6-fold increase in DF when 5% of PPP was added, resulting in a total fiber content of 4.82% in the bread. Tharsini & Sangwan (2018) [26] reported a 1.5-fold increase in DF when incorporating 6% of PPP in the bread formulation. Abolila et al. (2019)[21] described a substantial increase in DF (7.4-fold increase) upon fortifying bread with 18% PPP, reaching a DF content of 7.29% in the final product. However, regarding fiber, most of them determined crude fiber and did not measure DF, SDF, and IDF.

Overall, the acceptability of bread fortified with PPP was higher at addition levels that did not exceed 5%. For instance, Sayed-Ahmed (2014)[23] reported that bread fortified with 2.5% and 5% PPP received higher scores in overall acceptability and physical properties when compared to control bread and the 7.5% treatment. Similarly, based on sensory evaluation, Palak et al. (2020)[27] found that bread with 5% PPP treatment scored better than the 10% and 15% treatments in terms of color and appearance, body and texture, taste and flavor, and overall acceptability of the product. In parallel, the addition of PPP to bread can potentially have a substantial impact on the dough's characteristics. For instance, Abolila et al. (2019)[21] demonstrated that PPP-fortified loaves exhibited lower loaf volume compared to the control sample, with a reduction of 21.4% in bread containing 18% PPP.

Furthermore, there is scarce information regarding the glycemic response caused by the consumption of foods supplemented with PPP

3.- In a suitable section addressed to “Discussion” or “Future research directions”, I would introduce the future analysis of the market opportunities of such a bread, also considering its environmental (and economic) benefits. Nowadays, due to the inflation rate and the Russia-Ukraine conflict, the price of bread has steadily increased reaching never-experienced prices. Could the (partial) substation of wheat flour with pomegranate peel powder both an environmental and an economic strategy? Please, add some discussions on the topic.

Answer: In order to address the proposed topics, the following paragraph has been added Futures Prospects (L582-L609)

Consumer awareness of the environmental and health advantages related to bread products incorporating non-wheat ingredients, expected to exhibit a lower glycemic index, is on the rise. Recent global events, such as the COVID-19 pandemic and the Russian-Ukraine conflict, have had enduring impacts on wheat production and the supply chain[77,78].Thereby, the development of novel ingredients holds promise in mitigating pressure on food system resilience and bolstering food security. Diversifying plant-based food sources assumes paramount importance in enhancing sustainability within global food systems, as it not only reduces environmental impact but also fosters local economic development [79]. A diversified food supply chains facilitate access to affordable and nutritious food, particularly during crises, therefore promoting food system resilience. Additionally, non-wheat raw materials offer compelling nutritional properties, encompassing health-promoting compounds like dietary fibers and antioxidants [80]. Despite limitations in technological quality and sensory attributes relative to wheat substitution products, the transformation of food byproducts into valuable resources presents a significant opportunity to transition toward a closed-loop economic system[81]. Consumer decisions in the food market are influenced by various factors, including taste, smell, appearance, texture, functional properties, and nutritional value. Factors perceived as positively impacting health, such as product origin and food technology enhancing health benefits, also play a crucial role in consumer decision-making [82]. Considering the 2030 sustainable development goals, specifically goal 12 related to responsible production and consumption, addressing the existing knowledge gaps in nutritional, health, technological, and environmental dimensions through interdisciplinary approaches is imperative [83]. The synergy between human nutrition, food science, and sustainability is indispensable for scaling up research outcomes and achieving substantial positive impacts on a large scale[84]. Future research could be aimed at expanding the area of study and/or the limits of the system which innovations can reduce the costs of inputs, also involving other stages of the supply chain, or evaluate consumer acceptance in case of modifications of the recipe [78].

4.- In the subsection “Calculation of glycemic index of bread”, how the nine healthy adults have

been selected? Which sampling strategy was adopted? Please, add some more information. It nine adults a representative sample? I suggest the authors highlighting such biases/limitations in the research.  How the nine healthy adults have been selected? Which sampling strategy was adopted?

Answer: the subjects were selected according to the following protocol and that is indicated in the

Text (L225-L231): “Nine healthy adults (six women, three men) between 27 and 44 years old, with a body mass

index (BMI) between 18.5 and 24.9 kg/m², who had maintained their body weight for the last six months, were selected. Anthropometric evaluations, including weight and height, were conducted

using standardized methods. Only healthy individuals who did not consume any medication or supplements were considered. Those with diagnosed underlying pathologies, allergies or intolerances to test foods, and women with polycystic ovary syndrome were excluded from the study”.

It nine adults a representative sample?

Answer: According to Food and drug organization (FAO) the protocol to determine the GI of the food should be repeated at least in more than six subjects and the resulting GI values averaged. Normally, the GI for more than one food would be determined in one series of tests, for example, each subject might test two foods once each and the standard food once time for a total of three tests in random order on separate days. Subjects are studied on separate days in the morning after a 10-12 h overnight fast. For this reason, this is a representative sample according to this protocol.

I suggest the authors highlighting such biases/limitations in the research. 

Answer: L560-L562 In text was added: “Among the weaknesses of the study we can mention that the insulin response or the insulinemic index were not analyzed. In addition, we did not assess the menstrual phase of the volunteers, which may affect insulin sensitivity”.

5.- you should make reference to studies analyzing the bread supply chain, as to open new paths to environmental and economic studies. In my opinion, it is important to highlight the nexus between eco-friendly friendly and healthy products, also in the direction of the Sustainable Development Goals of the United Nations.

Answer: L66-L69. In text was added. In this context, the 2030 agenda for the United Nations sustainable development goals (SDGs) set food waste reduction targets (SDG 12). Due to this, several countries have adopted strategies to move toward a circular economy.

Answer: In order to address the proposed topics, the following paragraph has been added Futures Prospects (L582-L609)

Consumer awareness of the environmental and health advantages related to bread products incorporating non-wheat ingredients, expected to exhibit a lower glycemic index, is on the rise. Recent global events, such as the COVID-19 pandemic and the Russian-Ukraine conflict, have had enduring impacts on wheat production and the supply chain[77,78].Thereby, the development of novel ingredients holds promise in mitigating pressure on food system resilience and bolstering food security. Diversifying plant-based food sources assumes paramount importance in enhancing sustainability within global food systems, as it not only reduces environmental impact but also fosters local economic development [79]. A diversified food supply chains facilitate access to affordable and nutritious food, particularly during crises, therefore promoting food system resilience. Additionally, non-wheat raw materials offer compelling nutritional properties, encompassing health-promoting compounds like dietary fibers and antioxidants [80]. Despite limitations in technological quality and sensory attributes relative to wheat substitution products, the transformation of food byproducts into valuable resources presents a significant opportunity to transition toward a closed-loop economic system[81]. Consumer decisions in the food market are influenced by various factors, including taste, smell, appearance, texture, functional properties, and nutritional value. Factors perceived as positively impacting health, such as product origin and food technology enhancing health benefits, also play a crucial role in consumer decision-making [82]. Considering the 2030 sustainable development goals, specifically goal 12 related to responsible production and consumption, addressing the existing knowledge gaps in nutritional, health, technological, and environmental dimensions through interdisciplinary approaches is imperative [83]. The synergy between human nutrition, food science, and sustainability is indispensable for scaling up research outcomes and achieving substantial positive impacts on a large scale[84]. Future research could be aimed at expanding the area of study and/or the limits of the system which innovations can reduce the costs of inputs, also involving other stages of the supply chain, or evaluate consumer acceptance in case of modifications of the recipe [78].
